# TSDINO: Teacher-Student Self-Distillation Framework for Robust Pre-Training of Time-Series Foundation Models

## Abstract

Building time-series foundation models (TSFM) poses challenges in terms of learning stability due to limited data availability and heterogeneous temporal dynamics across various time-series datasets. We propose TSDINO, a teacher-student framework for robust pre-training of TSFM based on the principle of self-distillation with no labels. TSDINO offers a model-agnostic approach that combines two complementary objectives: (i) feature preservation under augmentations and (ii) masked patch prediction. A meta-architecture comprising teacher-student networks and projection heads enables adaptation to various models. We evaluate TSDINO on classification and forecasting tasks using diverse publicly available benchmarking datasets. TSDINO consistently achieves competitive zero-shot performance over gradient-based pre-training.

## 1 Introduction

Time series data are ubiquitous and at the operational core of modern sensing and decision-making systems over finance, energy, climate, healthcare, and the web. *Time-Series Foundation Models* (TSFM) (Liang et al., 2024; Das et al., 2024) offer a compelling paradigm by enabling zero-shot or few-shot generalization across domains, tasks, and diverse conditions in time-series analysis. This capability significantly reduces the cost, lead time, and engineering effort required to deploy systems and services that rely on time-series data.

Compared to language and vision, where foundation models are well established (Achiam et al., 2023; Liu et al., 2024a; Oquab et al., 2024), time series present unique challenges. Variability in domains and acquisition conditions, such as sampling rates, noise levels, and missing data patterns, introduces highly heterogeneous temporal dynamics. At the same time, training data is often scarce, noisy, or impractical to obtain at scale. These factors make it challenging to train TSFM that produce representations that transfer stably and robustly across tasks, regardless of domain or condition diversity.

While the development of TSFM has accelerated, most research has concentrated on architectural innovations rather than training methodologies. Chronos (Ansari et al., 2024) and TimesFM (Das et al., 2024) adopt a Transformer-based architecture, Moirai (Liu et al., 2024b) introduces any-variate attention to take attention over arbitrary numbers of variables, and TinyTimeMixer (TTM) (Ekambaram et al., 2024) and TSPulse (Ekambaram et al., 2025) explore lightweight MLP-Mixer-style designs.

Despite these architectural advances, the refinement of training strategies for TSFM lags behind analogous progress in language and vision, and, basically, using ordinary gradient-based learning is dominant. Further exploring training paradigms is required that go beyond architecture, focusing on robust self-supervised objectives and stability under heterogeneous dynamics.

Teacher-student self-distillation framework, such as DINO (Caron et al., 2021), has demonstrated remarkable success in self-supervised pre-training for vision models, where the student network learns to align its predictions with those of a momentum-updated teacher under multiple augmented views of the same image. Unlike contrastive learning methods, the framework does not require large negative sets; thus, it scales efficiently and produces semantically meaningful clusters, enabling

strong and stable transfer to diverse downstream tasks. Masked image modeling approaches (Zhou et al., 2022b) were also introduced to the framework as patch-level prediction objectives to capture local structures, improving robustness. Oquab et al. (2024) highlights the importance of multi-scale objectives.

In the time-series domain, such teacher-student framework has recently gained attention. Pieper et al. (2023) adapted data2vec-style distillation for time series, while Gao et al. (2024a) explored distillation-enhanced forecasting with momentum contrastive learning. However, these methods target task-specific models requiring supervised learning or fine-tuning and have not been extended to pre-training of TSFMs enabling zero-shot prediction. Moreover, existing approaches often emphasize global invariances but underrepresent local temporal patterns addressed in vision (Zhou et al., 2022b; Oquab et al., 2024).

To address these challenges, we propose **TSDINO**, a *Time-Series-tailored teacher-student framework for robust self-DIstillation with NO labels*. The student is trained to match its outputs with the teacher's outputs as self-distillation, and the teacher is an exponential moving average (EMA) of the past student, providing a stable target for the student. The matching is conducted with two complementary objectives: (i) *sequence-level feature preservation* under augmentations for global invariance, and (ii) *masked patch prediction* for local structure modeling. Within this framework, while the student processes augmented and masked input compared to the teacher's input, we propose augmentation and masking to be specialized for time series. The proposed meta-architecture of TSDINO enables adaptation to various models (e.g., TSPulse (Ekambaram et al., 2025) and TTM (Ekambaram et al., 2024)) and tasks (e.g., classification and forecasting). TSDINO provides a robust, stable, scalable, and model-agnostic pre-training framework for TSFMs.

**Contributions.**

- We present **TSDINO**, a novel model-agnostic teacher-student self-distillation framework for pre-training of TSFMS. TSDINO unifies sequence-level feature preservation with patch-level masked prediction, enabling robust representations for diverse downstream tasks such as classification and forecasting.

- We propose augmentation and masking strategies tailored for time series, departing from standard vision practices, to stabilize learning from time series.

- We demonstrate TSDINO's effectiveness across multiple tasks, classification, and forecasting. Our results indicate consistent zero-shot improvements over gradient-based pre-training, and ablations confirm the critical role of the unified objective in considering both global and local invariance.

## 2 RELATED WORK

**Self-supervised learning for time series.** Self-supervised learning (SSL) for time series spans contrastive, generative (e.g., masked modeling), and adversarial paradigms, each with distinct assumptions and augmentation needs (Zhang et al., 2024). Recent works explored self-distillation for time series: Pieper et al. (2023) adopted data2vec-style objectives, Gao et al. (2024a) combined distillation with contrastive learning, and similarity distillation (Hajimoradlou et al., 2022) leveraged pairwise similarities. However, these approaches lack zero-shot capabilities. TSDINO addresses these gaps to enable zero-shot prediction and robust transfer across tasks by unifying sequence-level and patch-level self-distillation.

**Self-Distillation in Vision.** Teacher-student self-distillation has become a cornerstone of self-supervised learning in vision. DINO (Caron et al., 2021) introduced a non-contrastive approach that aligns student predictions with a momentum teacher across augmented views. iBOT (Zhou et al., 2022b) and DINOv2 (Oquab et al., 2024) extended this paradigm by incorporating masked image modeling, enabling multi-scale representation learning. These methods highlight the importance of teacher-student framework and multi-scale objectives for robust feature learning. TSDINO adapts these principles to time series, addressing unique challenges such as temporal continuity, frequency sensitivity, and heterogeneous dynamics.

**Time-series foundation models.** Liang et al. (2024) provides surveys systematizing architectures of TSFMs. Inspired by large language models, decoder-only models such as TimesFM (Das et al., 2024) and Chronos (Liang et al., 2024) have demonstrated the feasibility of zero-shot forecasting. Other architectures, including MLP-Mixer-style designs (TTM (Ekambaram et al., 2024) and TSPulse (Ekambaram et al., 2025)), have also been explored. However, most efforts focus on architectural innovations rather than training methodologies. TSDINO contributes a robust, model-agnostic self-distillation framework that complements these models, enhancing their ability to learn robust representations across tasks and domains.

**Tasks in Time-Series Analysis and Modeling.** Time-series analysis encompasses tasks such as classification, clustering, retrieval, and forecasting, driving applications in finance, energy, healthcare, and beyond. All of the tasks benefit from robust, transferable representations. TSFMs provide *zero-shot* or few-shot generalization across domains and frequencies in the tasks (Ekambaram et al., 2024; 2025). TSDINO introduces a teacher-student self-distillation pre-training strategy that generally handles multiple models and tasks.

## 3 SELF-SUPERVISED PRE-TRAINING FOR TIME SERIES FOUNDATION MODELS

Our goal is to construct a self-supervised learning algorithm for *Time-Series Foundation Models* (TSFM) that, given unlabeled time series, learns (i) an encoder for general-purpose representation of time-series and (ii) a zero-shot forecaster that predicts future values in time-series without task-specific fine-tuning. The framework is model-agnostic and compatible with various archtectures (TSPulse, TTM etc.) and downstream tasks (classification and forecasting).

### 3.1 SELF-SUPERVISED PRE-TRAINING FRAMEWORK FOR TIME SERIES

Let $\mathcal{X} = \{\mathbf{x}_i\}_{i=1}^N$ be $N$ unlabeled time series, where each sample $\mathbf{x}_i \in \mathbb{R}^{P \times \tau}$ is a patchfied time series having $L = P \times \tau$ time steps, where each sequence is partitioned into $P$ non-overlapping patches having $\tau$ consecutive steps, following Nie et al. (2023).

Our objective is to learn a prediction model $f_{\boldsymbol{\theta}} : \mathbb{R}^{P \times \tau} \to \mathbb{R}^d$ that maps a time series to output for each task $\mathbf{z}_i = f_{\boldsymbol{\theta}}(\mathbf{x}_i)$ described in the subsequent subsection. The model is parameterized by $\boldsymbol{\theta}$.

### 3.2 TASK-SPECIFIC OBJECTIVES

**(1) General-purpose representation learning.** The model $f_{\boldsymbol{\theta}}$ is an encoder that maps a time series $\mathbf{x}_i$ to a feature vector $\mathbf{z}_i$ such that $\text{similarity}(\mathbf{z}_i, \mathbf{z}_j) \approx \text{semantic similarity}(\mathbf{x}_i, \mathbf{x}_j)$, enabling downstream tasks such as clustering, retrieval, and classification without supervision.

**(2) Zero-shot forecasting.** The model $f_{\boldsymbol{\theta}}$ is a forecaster that given $\mathbf{x}_i$ with time steps $1 : L$, predict future values $\mathbf{z}_i = \hat{\mathbf{x}}_i$ with time steps $L + 1 : L + 1 + d$ without task-specific fine-tuning.

## 4 TEACHER-STUDENT FRAMEWORK FOR PRE-TRAINING OF TIME SERIES FOUNDATION MODELS

We address self-supervised pre-training for TSFMs via a teacher-student framework, where we use two different networks, student $f_{\boldsymbol{\theta}_S}$ and teacher $f_{\boldsymbol{\theta}_T}$, which share the same architecture but have separate parameters $\boldsymbol{\theta}_S$ and $\boldsymbol{\theta}_T$ to learn from unlabeled time series.

Overall, the student parameter is trained via gradient descent to match the student's outputs with the teacher's outputs as self-distillation, where the teacher is an exponential moving average (EMA) of the past parameter sequence of the student parameters. The matching between the student and the teacher outputs is conducted at the sequence level (feature preservation under augmentations) and the patch level (masked patch prediction), which encourages the model to learn both global invariances and local temporal structures. The momentum teacher provides stable targets for the student, and centering and sharpening of the teacher's outputs further stabilize training.

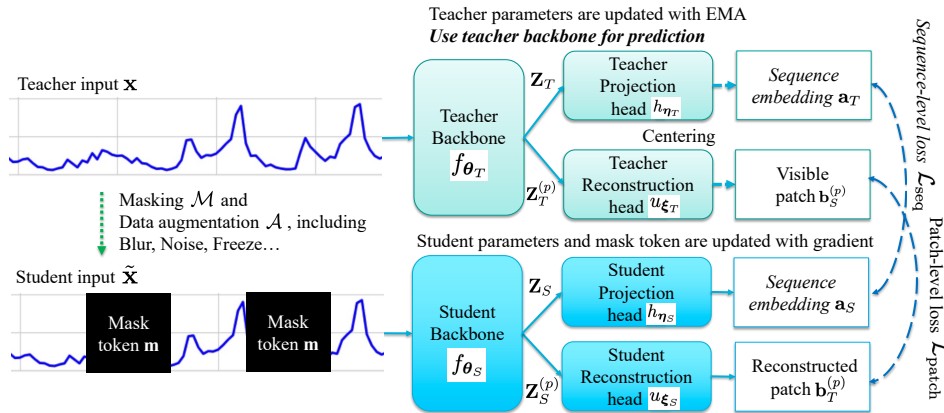

Figure 1: TSDINO training pipeline.

This framework encourages the student to learn robust, invariant representations that generalize across domains and tasks, and the teacher reaches a more stable and robust variant of the student; thus, we finally use the teacher network for prediction.

### 4.1 OVERVIEW OF TRAINING PIPELINE.

The training pipeline is illustrated in Figure 1.

First, for an input of the student, we generate a binary mask indicator $\mathbf{I}_i \in \{0,1\}^P$ to each patch randomly, and an augmented and masked view is created:

$$\tilde{\mathbf{x}} \equiv \mathcal{M}\big(\mathcal{A}(\mathbf{x}), \mathbf{I}\big), \tag{1}$$

where $\mathcal{A}$ is time-series augmentation and $\mathcal{M}$ perform patch-wise value-fill with the mask indicator $\mathbf{I}$ (further details are in §4.4). An input for the teacher is the original input sequence, in contrast to the vision literature, using the augmentation for both the student and the teacher (Caron et al., 2021; Zhou et al., 2022a) since the uncertainty of time-series data is much larger than that of image data, and strong augmentations may change the semantics of the time series.

The teacher and student respectively process the original input $\mathbf{x}$ and the student input $\tilde{\mathbf{x}}$ to produce embeddings of all $P$ patches, $\{\mathbf{Z}_T, \mathbf{Z}_S\} \in \mathbb{R}^{P \times d}$, as

$$\mathbf{Z}_T \equiv f_{\boldsymbol{\theta}_T}(\mathbf{x}) \quad \text{and} \quad \mathbf{Z}_S \equiv f_{\boldsymbol{\theta}_S}(\tilde{\mathbf{x}}). \tag{2}$$

Then we compute projected sequence embeddings $\{\mathbf{a}_T, \mathbf{a}_S\} \in \mathbb{R}^C$ with the teacher and student as

$$\mathbf{a}_T \equiv h_{\boldsymbol{\eta}_T}(\mathbf{Z}_T) \quad \text{and} \quad \mathbf{a}_S \equiv h_{\boldsymbol{\eta}_S}(\mathbf{Z}_S), \tag{3}$$

where $h_{\boldsymbol{\eta}_T}$ and $h_{\boldsymbol{\eta}_S}$ are projection heads for teacher and student respectively. We also compute reconstructed patch embeddings $\{\mathbf{b}_T^{(p)}, \mathbf{b}_S^{(p)}\} \in \mathbb{R}^C$ for masked patches with patch indices $p \in \{p : I_p = 1, I_p \in \mathbf{I}\}$ with the teacher and student as

$$\mathbf{b}_T^{(p)} \equiv u_{\boldsymbol{\xi}_T}(\mathbf{Z}_T^{(p)}) \quad \text{and} \quad \mathbf{b}_S^{(p)} \equiv u_{\boldsymbol{\xi}_S}(\mathbf{Z}_S^{(p)}), \quad \text{where} \quad p \in I, \tag{4}$$

and $u_{\boldsymbol{\xi}_T}$ and $u_{\boldsymbol{\xi}_S}$ are reconstruction heads for teacher and student respectively.

From the outputs of the projection and reconstruction heads, we form the total loss $\mathcal{L}_{\text{SSL}}(\mathbf{a}_T, \mathbf{a}_S, \mathbf{b}_T, \mathbf{b}_S)$ and update the student parameters, $\boldsymbol{\Theta}_S \equiv \{\boldsymbol{\theta}_S, \boldsymbol{\eta}_S, \boldsymbol{\xi}_S\}$, by gradient descent on $\mathcal{L}_{\text{SSL}}$ as

$$\boldsymbol{\Theta}_S^{(s)} = \boldsymbol{\Theta}_S^{(s-1)} - \rho \nabla_{\boldsymbol{\Theta}_S} \mathcal{L}_{\text{SSL}}(\mathbf{a}_T, \mathbf{a}_S, \mathbf{b}_T, \mathbf{b}_S), \tag{5}$$

where $\rho$ is the learning rate. We add the forecasting loss when we use the trained model for forecasting, and the details of $\mathcal{L}_{\text{SSL}}$ are given in the subsequent subsection.

The teacher is not directly optimized by gradients but is maintained as an exponential moving average (EMA) of the student parameter sequence to provide stable targets. At step $s$ the teacher parameters, $\boldsymbol{\Theta}_T \equiv \{\boldsymbol{\theta}_T, \boldsymbol{\eta}_T, \boldsymbol{\xi}_T\}$, are updated as

$$\boldsymbol{\Theta}_T^{(s)} = m(s)\boldsymbol{\Theta}_T^{(s-1)} + (1 - m(s))\boldsymbol{\Theta}_S^{(s-1)}, \tag{6}$$

where the momentum $m(s)$ is typically scheduled to increase toward 1 over training (cosine schedule). EMA serves as a low-cost mechanism to stabilize self-supervised training and improve representation quality without introducing additional optimization complexity. EMA ensures that the teacher evolves slowly, producing consistent targets for the student and reducing training instability, and the teacher acts as a temporal ensemble of past student models, improving generalization and reducing variance, similar to Polyak averaging.

Empirically, prior work (Caron et al., 2021; Oquab et al., 2024) shows that EMA-based teachers consistently outperform their student counterparts on downstream tasks, validating its effectiveness. Therefore, at inference, we use the teacher network $f_{\boldsymbol{\theta}_T}$ for prediction, as it is a more stable and robust variant of the student.

## 4.2 Detailed Loss Functions for training Student.

We use cross-entropy (CE) loss for both sequence-level and patch-level objectives, following Caron et al. (2021); Zhou et al. (2022b); Oquab et al. (2024). The self-supervised objective for student parameters combines: $\mathcal{L}_{\text{SSL}} \equiv \mathcal{L}_{\text{seq}} + \alpha \mathcal{L}_{\text{patch}}$, where:

$$\mathcal{L}_{\text{seq}} \equiv \frac{1}{N} \sum_{i=1}^{N} \text{CE}\big(\{\mathbf{a}_T\}_i, \{\mathbf{a}_S\}_i\big), \tag{7}$$

$$\mathcal{L}_{\text{patch}} \equiv \frac{1}{\sum_{i=1}^{N} \sum_{p=1}^{P} I_{i,p}} \sum_{i=1}^{N} \sum_{p=1}^{P} I_{i,p} \text{CE}\big(\{\mathbf{b}_T\}_{i,p}, \{\mathbf{b}_S\}_{i,p}\big). \tag{8}$$

Here, $\alpha$ controls the contribution of masked prediction. We also add KoLeo regularization (Sablay-rolles et al., 2018; Oquab et al., 2024) to promote spreading of features and avoid collapse, improving embedding quality. The forecasting loss for the forecasting task is MSE, $\frac{1}{N} \sum_{i=1}^{N} |\mathbf{z}_i - \hat{\mathbf{x}}_i|^2$.

## 4.3 Model-agnostic Meta-architecture with Heads.

The proposed **TSDINO** adopts a unified and modulated model architecture for both of the teacher and student. They are sharing the same architecture but have independent parameters. Our model-agnostic meta-architecture consists of three main components: (i) backbone networks $f_{\boldsymbol{\theta}_T}, f_{\boldsymbol{\theta}_S}$ that extracts embeddings from input patches, (ii) projection heads $h_{\boldsymbol{\eta}_T}, h_{\boldsymbol{\eta}_S}$ that map embeddings to a normalized feature space for self-distillation, and (iii) reconstruction heads $u_{\boldsymbol{\xi}_T}, u_{\boldsymbol{\xi}_S}$ that reconstructs masked token targets. This design is agnostic to the specific model choice (e.g., TTM, TSPulse) as it is backbone for the meta model and supports various downstream tasks (classification and forecasting). The teacher further has centering and sharpening components to stabilize training.

**Projection and Reconstruction Heads.** Projection heads $h_{\boldsymbol{\eta}_T}, h_{\boldsymbol{\eta}_S} : \mathbb{R}^{P \times d} \to \mathbb{R}^C$ are three-layer MLPs with GELU activation, normalization, and a softmax layer at the top layer and have student and teacher concatenated embeddings over patches after the backbone as their inputs. Reconstruction heads $u_{\boldsymbol{\xi}_T}, u_{\boldsymbol{\xi}_S} : \mathbb{R}^d \to \mathbb{R}^C$ are also three-layer MLPs with GELU activation, normalization, and a softmax layer at the top layer, and have student and teacher patch embeddings after the backbone as their inputs. The heads for teachers $h_{\boldsymbol{\eta}_T}, u_{\boldsymbol{\xi}_T}$ further have a centering and sharpening component, which centers the teacher outputs and sharpens before computing the distillation loss to prevent representation collapse and stabilize training (Oquab et al., 2024).

For forecasting, we utilize the decoder from Tiny-Tim-Mixers (TTM) (Ekambaram et al., 2024).

## 4.4 Augmentation techniques and Masking strategies for Time Series

The teacher network always receives the original sequence $x$, while the student network processes the augmented and masked view $\tilde{x}$. This asymmetric design stabilizes training by providing consistent

targets while allowing the student to learn invariances under perturbations. For student input, we apply data augmentation $\mathcal{A}$ and patch-wise temporal masking $\mathcal{M}$.

**Data Augmentation for Student.** We define an augmentation operator $\mathcal{A}$ that stochastically applies a set of transformations: $\mathcal{A}(x) = \mathcal{T}_k \circ \cdots \circ \mathcal{T}_1(x)$, where $\{\mathcal{T}_i\}$ are sampled from a pool of augmentations. Specifically, we use Gaussian noise: $\mathcal{T}(x) = x + \epsilon$, where $\epsilon \sim \mathcal{N}(0, \sigma^2)$, Salt-and-pepper noise: randomly replace points with $\min(x)$ or $\max(x)$ with probability $p$, Gaussian blur: apply convolution with a Gaussian kernel to smooth high-frequency components, FreezeSection: randomly select a consecutive segment $[j, k]$ and set $x_{j:k} = x_j$, and Random time shift circularly shift the sequence by step $\delta$.

**Patch-wise temporal masking.** We use patch-wise temporal masks to encourage capturing local temporal structures. Let $\{\mathbf{x}^{(p)}\}_{p=1}^P$ be the sequence of patches, each $\mathbf{x}^{(p)} \in \mathbb{R}^\tau$. We sample patch index $p$ until a target mask ratio $r_{\text{mask}} \in (0, 1)$ is reached, yielding the masked and unmasked index set for student $I$. We replace masked patches with a single learnable embedding (mask token) $\mathbf{m} \in \mathbb{R}^\tau$ to simplify implementation and provide an explicit signal of missingness. The mask token is also updated with student update rule (gradient): $\mathbf{m}^{(s)} = \mathbf{m}^{(s-1)} - \rho \nabla_{\mathbf{m}} \mathcal{L}_{\text{SSL}}(\mathbf{a}_T, \mathbf{a}_S, \mathbf{b}_T, \mathbf{b}_S)$.

## 5 EXPERIMENTS

**Scope of Experiments.** The goal of this paper is *to introduce a novel pre-training framework for TSFMs* and demonstrate its effectiveness, *not to propose a new TSFM*. Specifically, we aim to show that the proposed method can maximize the performance of existing TSFM architectures in zero-shot scenarios. Therefore, for each task, we adopt an existing TSFM as the backbone network in TSDINO and replace only the pre-training procedure from gradient-based pre-training to TSDINO, ensuring that any performance gains are attributable to the learning method rather than architectural changes. Consequently, the only difference between our method and the baseline lies in the pre-training procedure, while the model architecture remains identical.

We first describe the pre-training setup and show evaluation results of **TSDINO** on primary tasks for TSFMs in zero-shot scenarios: (i) *classification* and (ii) *forecasting*. We follow standard protocols used in prior work on forecasting and representation learning benchmarks (Zhou et al., 2021; Wu et al., 2021; Zeng et al., 2023; Nie et al., 2023; Ekambaram et al., 2024; 2025; Bagnall et al., 2018).

**Pre-training Setup and Training Dataset.** TSDINO pre-training utilizes a subset of 1B time points drawn from the combined dataset of the Monash (Godahewa et al., 2021) and LibCity (Wang et al., 2021; Woo et al., 2024) data collections. We leverage the same pre-training data as used in TTM (Ekambaram et al., 2024) and TSPulse (Ekambaram et al., 2025), following their data selection and pre-processing procedures. All datasets used are selected under permissive licenses that allow both open-source and commercial use. We first pretrain the model using the objectives in §4.2 with the dataset. After pre-training, we *freeze* the backbone network weights and evaluate in zero-shot downstream tasks. Pre-training uses a single $\times$A100-80GB GPU, which took around 12 hours.

### 5.1 CLASSIFICATION

Time-series classification is a fundamental task in time-series analysis. Our objective is to evaluate zero-shot performance without task-specific fine-tuning. For this purpose, we consider two settings: (i) similarity search and retrieval (§5.1.1) and (ii) classification with a frozen backbone (§5.1.2). Similarity search and retrieval finds the time series from a single reference example and is a pure zero-shot task, which can be seen as zero-shot classification. Classification with a frozen backbone is not a pure zero-shot task but is a common evaluation protocol in representation learning; thus, we will investigate it for reference.

**Baselines** We use TSPulse (Ekambaram et al., 2025) as the backbone network for both the student and teacher networks of TSDINO due to its strong performance in classification and publicly available implementation, which enables us to conduct local pre-training. We compared `TSDINO` with TSPulse trained using the original gradient-based pre-training, referred to as `Gradient`. Note that TSDINO and `Gradient` share the same model architecture, differing only in the pre-training strategy. Also,

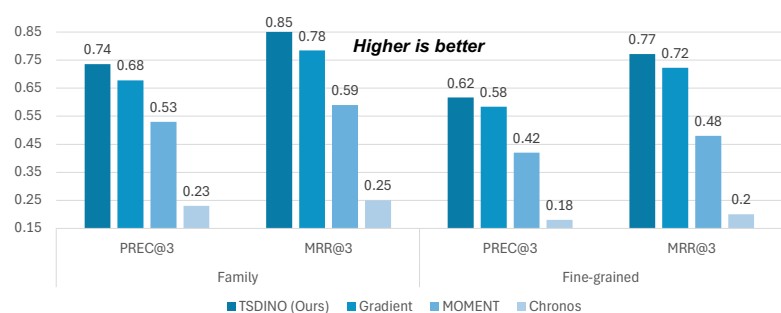

Figure 2: Zero-shot search / retrieval results on UCR dataset. The figure shows PREC@K and MRR@K (K = 3) for `TSDINO` compared to baselines (`Gradient`, `Chronos`, `MOMENT`).

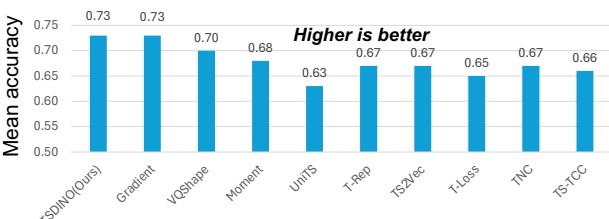

Figure 3: Classification accuracy on UEA datasets.

TSPulse has a single version, making reproducibility straightforward, which also allows us to completely follow the comparison with the compared methods in the paper. Following the paper, we use: `Chronos` (Ansari et al., 2024), and `MOMENT` (Goswami et al., 2024) in §5.1.1. In §5.1.2, we compare with **Data-specific models.** T-Rep (Fraikin et al., 2023), textitTS2Vec (Yue et al., 2022), T-Loss (Franceschi et al., 2019), TS-TCC (Eldele et al., 2021), `TNC` (Tonekaboni et al., 2021). **Zero-shot forecasters (TSFMs, reference only).** `TSPulse` (Ekambaram et al., 2025), **VQShape** (Wen et al., 2024), `MOMENT` (Goswami et al., 2024), and `UniTS` (Gao et al., 2024b).

### 5.1.1 TIME-SERIES SIMILARITY SEARCH & RETRIEVAL AS ZERO-SHOT CLASSIFICATION

**Setup** We evaluate time-series similarity search & retrieval as zero-shot classification, where we find the time series from a single reference example. We use a synthetic dataset and a real dataset based on the UCR dataset (Dau et al., 2019), following the setting in (Ekambaram et al., 2025). After pre-training, we freeze the encoder, and retrieval is performed using cosine similarity of embeddings. We report PREC@K and MRR@K (K = 3). Higher values indicate better performance. No fine-tuning or additional adaptation is applied.

**Results** Figure 2 shows retrieval performance. TSDINO consistently improves the gradient-based pre-training and outperforms the baselines, demonstrating that the proposed method yields representations that better capture semantic similarity.

### 5.1.2 FINE-TUNED CLASSIFICATION WITH FROZEN BACKBONE

**Setup** For reference, while it is not zero-shot task, we conducted evaluation on fine-tuned classification, following the setting in (Ekambaram et al., 2025). We evaluate on 29 datasets from the UEA Multivariate Time Series Classification Archive (Bagnall et al., 2018). Standard train/test splits are used. We report classification mean accuracy over classes. Higher values indicate better performance.

**Results** As shown in Fig. 3, `TSDINO` achieves higher accuracy than most of the baselines and performs comparably to `Gradient`. Fig. 4 visualizes results across datasets. Among 29 datasets, `TSDINO` improves `Gradient` in 16 datasets, indicating that their overall performance is comparable, but `TSDINO` yields a different local optimum of the weights from gradient-based pre-training.

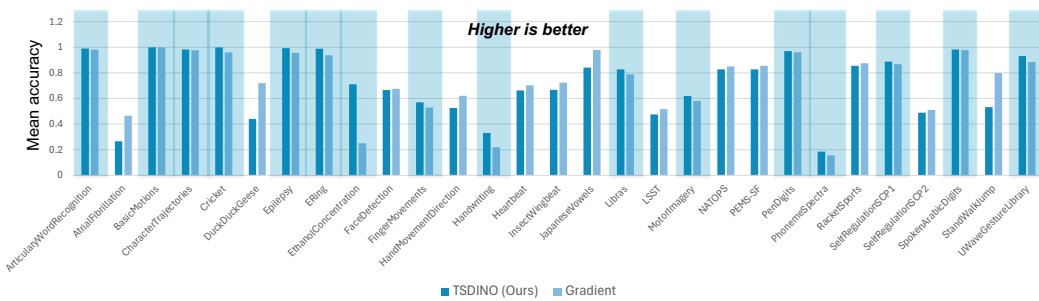

Figure 4: Per-dataset classification accuracy on UEA datasets.

Table 1: Comparison of MSE in zero-shot forecasting (best in **bold**, second-best underlined).

| Dataset | TSDINO (Ours) | Gradient | TTM (From paper, For ref.) |
|---|---|---|---|
| ETTH1 | 0.399 | 0.404 | **0.395** |
| ETTH2 | **0.337** | 0.346 | 0.345 |
| ETTM1 | **0.382** | 0.410 | 0.386 |
| ETTM2 | **0.278** | 0.305 | 0.281 |
| Weather | 0.239 | 0.260 | **0.237** |
| Electricity | 0.221 | 0.231 | **0.205** |
| Traffic | 0.567 | 0.579 | **0.564** |

## 5.2 FORECASTING

Time-series forecasting is also a core task in time-series analysis. We evaluate TSDINO's ability to perform zero-shot forecasting on unseen datasets without task-specific fine-tuning.

**Baselines** We use TTM (Ekambaram et al., 2024) as the backbone network for both the student and teacher networks of TSDINO due to its strong performance in zero-shot forecasting and publicly available implementation, which enables us to conduct local pre-training. We compared TSDINO with TTM trained using the original gradient-based pre-training, referred to as Gradient. Note that TSDINO and Gradient share the same model architecture, differing only in the pre-training strategy. TTM has multiple versions, making reproducibility challenging; therefore, we report both our reproduced results Gradient and the reference values from the original paper, referred to as TTM (From paper, For ref.).

**Setup** We evaluate zero-shot forecasting on widely used multivariate benchmarks: *ETTh1*, *ETTh2*, *ETTm1*, *ETTm2*, *Electricity*, and *Traffic* (Zhou et al., 2021; Nie et al., 2023). These datasets cover diverse temporal patterns and scales, including strong seasonality (Electricity), high-dimensional traffic flows, and energy-related series with varying granularity. We follow standard preprocessing and splits from prior work (Nie et al., 2023; Wu et al., 2021; Zeng et al., 2023; Ekambaram et al., 2023), and evaluate multiple forecast horizons $H \in \{96, 192, 336, 720\}$. We use Mean Squared Error (MSE) as evaluation metrics, reporting average performance across all forecast horizons. Lower values indicate better performance.

**Results** Table 1 summarizes zero-shot forecasting performance. TSDINO consistently improves Gradient across all datasets, demonstrating the effectiveness of the proposed pre-training method. Compared to TTM (From paper, For ref.), TSDINO achieves competitive performance.

## 5.3 ABLATION STUDY

### 5.3.1 LEARNING OBJECTIVES

We measure the contribution of loss components in §4.2, TSDINO (Ours): Sequence-level loss + Patch-level loss, Sequence: Sequence-level loss only, and Patch: Patch-level loss only, in terms of

Table 2: Ablation study: For ZS classification, we report PREC@3 and MRR@3 (higher is better). For For FT classification, we report mean accuracy (higher is better). For forecasting, we report MSE (lower is better). Best in **bold** and second-best underlined.

| Task | Teacher-student EMA | | | Gradient |
|---|---|---|---|---|
| | TSDINO (Ours) | Sequence | Patch | |
| ZS Classification (Family, PREC@3) | **0.735** | 0.700 | 0.701 | 0.678 |
| ZS Classification (Family, MRR@3) | **0.850** | 0.822 | 0.814 | 0.784 |
| ZS Classification (Fine-grained, PREC@3) | **0.616** | 0.580 | 0.571 | 0.584 |
| ZS Classification (Fine-grained, MRR@3) | **0.771** | 0.737 | 0.718 | 0.723 |
| FT Classification (Mean accuracy) | **0.730** | 0.707 | 0.713 | **0.730** |
| Forecasting (MSE, Lower is better) | **0.346** | 0.358 | 0.358 | 0.362 |

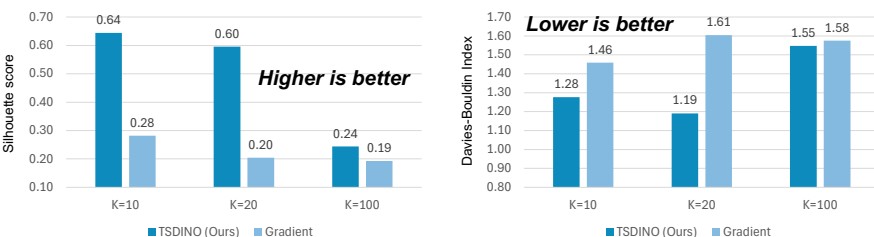

Figure 5: Clustering metrics comparison using $k$-means on frozen features ($k = 10, 20, 100$). TSDINO improves the gradient-based pre-training, Gradient, in silhouette score and Davies-Bouldin index, indicating better clustering quality.

performance improvement from the gradient-based pre-training in tasks, zero-shot (ZS) classification in §5.1.1, fine-tuned (FT) classification in §5.1.2, and forecasting in §5.2, as shown in Table 2. Both sequence-level and patch-level losses contribute to performance gains, with the full combination yielding the best results overall. This indicates that learning both global invariances and local temporal structures is crucial for effective time-series representation learning. The teacher-student framework and EMA weight updates basically contribute to improving performance compared to the ordinary gradient-based training.

### 5.3.2 CLUSTERING RESULTS OF EMBEDDINGS

To investigate the quality of learned representations, we evaluate clustering performance on embeddings. We apply $k$-means on embeddings ($k = \#$classes) and compute the silhouette score and Davies-Bouldin Index. A higher silhouette score and lower Davies-Bouldin index indicate better clustering quality. We compare TSDINO with the gradient-based pre-training, Gradient, using TSPulse.

**Results** TSDINO improved clustering metrics compared to Gradient by a good margin, indicating that its representations effectively capture semantic similarity. This nature aligns with good zero-shot performances in various downstream tasks.

## 6 CONCLUSION

We presented TSDINO, a self-supervised pre-training framework for time-series foundation models based on the teacher-student self-distillation framework. Our design incorporates time-series augmentation and patch-wise masking to respect temporal continuity and structure. Extensive experiments demonstrate that TSDINO learns robust, generalizable features that excel in zero-shot tasks, including classification and forecasting across diverse benchmark datasets. Ablations confirm the importance of each component in our objectives.

**Reproducibility statement**    We wrote the details of experimental settings, computing resources, and datasets in §5. We plan to release the code after the paper is accepted.

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
