# OpenReview forum: "TSDINO: Teacher–Student Self-Distillation Framework for Robust Pre-training of Time-Series Foundation Models"
_ICLR.cc/2026/Conference — Submitted to ICLR 2026_

### Official Review · Reviewer_aesc · 2025-10-27

**Soundness:** 2
**Presentation:** 2
**Contribution:** 2
**Rating:** 2
**Confidence:** 4

**Summary:**

In this paper the authors propose a distillation based pretraining paradigm for time series foundation models, in particular TSFM encoders. They suggest that the teacher uses the exponential average of students' weights, while the students match the teacher's hidden representations on both the patch and the sequence level using obfuscated inputs. In the empirical study section the authors conclude that such training paradigm yields a better pretrained encoder compared to standard SGD when using TSPulse as the backbone.

**Strengths:**

They paper introduces a novel and legit research question regarding the best training paradigm for TSFM. The proposed method is theoretically sound.

**Weaknesses:**

The conclusions made in the paper are not well supported. There are questions in both the theoretical and the empirical perspectives that need to be addressed to make the conclusions credible. See questions.

The writing of the paper is a bit ambiguous and is lacking key details.

**Questions:**

Regarding theory:

1. To be nitpicky, with the focus actually on the teacher model, the proposed paradigm is more aligned with meta learning than (self?)-distillation, that the whole teacher student interaction seems equivalent to an adaptive learning rate scheduler with momentum. Can you provide some theoretical justification (high level is fine) on why the proposed method is superior over modern SGD?

2. Can yoy justify the assumption that the chosen augmentations for the student's inputs should not affect the student's hidden representation of such inputs?

3. Minor: how is the ground truth label factored in the students' loss?

Regarding the empirical study:

1. Can you provide the training dynamics of TSDINO TSPulse and Gradient TSPulse, or alternatively a notion of "being equally trained" between the two models.  For example suppose both approaches are trained with the same steps then the distillation approach uses more FLOPS. An ideal evidence here is that both model trainings have converged. Without such evidence it is unclear whether the edge of TSDINO TSPulse comes from a better training paradigm or from more training.


2. If the claim is that paradigm is model class agnostic, then another backbone model class should be involved to at least attempt to test the generalizability of such claim.

3. Figure 4 is lacking evidence on statistical significance.

4. Minor: for zero shot forecasting, the mentioned 7 datasets are limiting and overused. The practice of using a window of 96 to forecast at most 720 points is not sound. Consider a slightly more diversified and comprehensive benchmark.

---

> ### Author Response · Authors · 2025-12-04
> **Response to Reviewer aesc**
>
> Thank you for the helpful comments and suggestions.
>
> > Minor: how is the ground truth label factored in the students' loss?
>
> For general-purpose representation learning. with TSPulse, the ground truth labels are not directly factored into the student's loss during pre-training. The focus is on aligning the student's representations with those of the teacher through self-supervised objectives. For forecasting with TTM, the forecasting loss is added from the TTM original loss, the Mean Squared Error (MSE) between the predicted future values and the actual future values.
>
> > Can you provide the training dynamics of TSDINO TSPulse and Gradient TSPulse, or alternatively a notion of "being equally trained" between the two models. For example suppose both approaches are trained with the same steps then the distillation approach uses more FLOPS. An ideal evidence here is that both model trainings have converged. Without such evidence it is unclear whether the edge of TSDINO TSPulse comes from a better training paradigm or from more training.
>
> TSDINO flops and training steps are almost aligned with those of Gradient TSPulse. We train only the student network with gradients. The teacher network's parameters are updated using an EMA of the student parameters, which incurs negligible computational cost.
>
> > Minor: for zero shot forecasting, the mentioned 7 datasets are limiting and overused. The practice of using a window of 96 to forecast at most 720 points is not sound. Consider a slightly more diversified and comprehensive benchmark.
>
> Forecasting at most 720 points from a 512 context horizon is the setting in the original TTM paper, which should be reasonable. We will consider more diversified and comprehensive benchmarks in the future.

---

### Official Review · Reviewer_umYb · 2025-10-29

**Soundness:** 2
**Presentation:** 2
**Contribution:** 1
**Rating:** 2
**Confidence:** 3

**Summary:**

This paper proposes a teacher-student self-distillation framework (TSDINO) for TSFM pre-training. TSDINO is an architecture-independent approach, meaning it can use different neural network architectures. Its training objective consists of feature preservation under augmentation plus masked patch prediction. The pre-trained model is suitable for prediction and classification tasks. The authors evaluated TSDINO on several publicly available benchmark datasets, and the experimental results validate its effectiveness.

**Strengths:**

The paper is written in a clear and easy-to-understand manner.

**Weaknesses:**

- The paper lacks originality. It directly uses the DINO method from the image domain to train the time series foundation model, without providing an analysis of the method's applicability to time series scenarios and evidence to explain why the method is effective.
- The paper's experiments are relatively weak, lacking validation on large-scale benchmarks.

**Questions:**

- A deep analysis to know why the DINO training strategy can work on time series data would be interesting.
- In the original DINO paper, specialized neural network architectures (such as VIT and ResNet) were required for representation extraction. Why can the authors claim here that TSDINO does not depend on the form of the network architecture?

---

> ### Author Response · Authors · 2025-12-04
> **Response to Reviewer umYb**
>
> Thank you for the helpful comments and suggestions.
>
> > In the original DINO paper, specialized neural network architectures (such as VIT and ResNet) were required for representation extraction. Why can the authors claim here that TSDINO does not depend on the form of the network architecture?
>
> The requirement of the proposed method is only the capability of extracting hidden states in the model for a sequence and each patch, which is usually possible for models based on neural networks (e.g., transformers, CNNs, RNNs). Therefore, the proposed method is model-agnostic and can be applied to various architectures.

---

### Official Review · Reviewer_WNzk · 2025-10-31

**Soundness:** 3
**Presentation:** 3
**Contribution:** 2
**Rating:** 4
**Confidence:** 4

**Summary:**

This paper proposes a DINO-like pre-training for TFMs based on a teacher-student distillation setup. The authors argue that such pre-training methods were not explored in the TSFM field before and suggest that they can work equally well in time series. The authors adapt DINO-based pre-training to time series and produce two models that follow such pre-training: one that distills TTM models for forecasting and another that distills the TSPulse model for discriminative tasks. Large-scale evaluations hint at the potential efficiency of this approach.

**Strengths:**

1.	A new pre-training approach adapted to time series from DINO pre-training.
2.	Variants of the model for discriminative and generative tasks.
3.	Promising performance in the considered tasks.

**Weaknesses:**

1.	Although distillation is not that explored in the TSFM field, different TSFMs have already explored other ways to pre-train models. Flow-based (FlowState) models based on SSM, diffusion-based models (Sundial), masked-based (MOMENT), etc. In classification, MANTIS-8M was pre-trained using contrastive learning, too, just as the self-supervised variant of UniTS. It is thus a bit of a stretch to claim that different pre-training strategies were not explored in TSFMs.
2.	TTMs are very far from being competitive in time series forecasting. It will be more instructive to distill into stronger models and compare with them (TimesFM 2.5 or TiRex).
3.	Same as above, Mantis-8M (Feofanov et al.) performs stronger in zero-shot than other FMs and was pre-trained contrastively. While I understand the argument that this is a proof-of-concept, I guess it still needs to be comparable to the best models for people to explore this idea further, not just be comparable to those that are far from the top of the leaderboard.

**Questions:**

1.	Is it possible to provide more evidence of the suitability of this approach for stronger forecasting/classification models? If such evidence is provided, I will increase my score to 8.
2.	Time series augmentations were explored in time series forecasting before (see TiRex paper), as well as patch masking. What other novelties do authors consider in this work that are specific to time series and/or novel?
3.	Is there any evidence of the pitfalls of the common pre-training approaches that DINO-like pre-training can solve in time series? It would be instructive to have motivational examples providing more intuition for why we expect the improvement with this kind of pre-training.

---

> ### Author Response · Authors · 2025-12-04
> **Response to Reviewer WNzk**
>
> Thank you for the helpful comments and suggestions. We will apply the proposed approach to a broader range of forecasting/classification models in our future work.

---

### Official Review · Reviewer_dBGd · 2025-11-01

**Soundness:** 3
**Presentation:** 1
**Contribution:** 2
**Rating:** 2
**Confidence:** 5

**Summary:**

This paper presents a self-supervised pre-training framework for Time-Series Foundation Models (TSFMs). The authors argue that existing TSFMs mostly focus on architectural innovations, rather than the training strategies. Therefore, TSDINO proposes to extend the teacher-student self-distillation paradigm originally popularized by DINO in computer vision to the time-series domain.

In TSDINO, there is a student network trained to match the outputs of a teacher network, which is maintained as an Exponential Moving Average (EMA) of the student. The training objective has two components: 1) sequence-level alignment, which aims to align the global representation of the selected time series window of the teacher and the student, and 2)  masked patch prediction, which aims to capture local temporal structures by predicting the representations of masked patches. The authors argue that a key design choice here is the asymmetric input strategy. The student processes augmented and masked data, while the teacher processes the original time series, as the authors attribute this to time series being more sensitive to noise.

TSDINO is designed to be model-agnostic. The evaluation involves integrating existing TSFM architectures as backbones for different tasks and comparing TSDINO against the original pre-training strategies of these models. The results show consistent improvements in zero-shot forecasting and zero-shot classification (retrieval), along with improved embedding quality as measured by clustering metrics.

**Strengths:**

1. The adaptation of the DINO framework to time series seems interesting.
2. The authors show good performance when compared to the limited baselines.

**Weaknesses:**

1. There are many writing and notation issues.
- Line 146: "The model fθ is a forecaster that given xi with time steps 1 : L, predict future values zi = ˆxi with time steps L + 1 : d without task-specific fine-tuning." --> Do you mean to write d here?
- Line 229: "Therefre, at inference, we use the teacher network fθT for prediction" --> "Therefore"
- Line 245: "Forecasting loss for the foerecasting taks is MAE" --> But then you showed the formula for MSE?
- In Equation 8, "\sum_{p=1\in P}" --> This is not proper notation.
- In Line 254, the projection and reconstruction heads are defined using parameters \eta and \xi (e.g., h_{\eta_{S}}). However, in Section 4.3, they are referred to using \theta (e.g., h_{\theta_{T}}). Notation must be consistent.
- What is the dimensionality of a_T, a_S in the sequence embeddings?
- Line 215: "forecasiting" -> "forecasting".
- Line 190: "where A is time-series augmentation and A perform patch-wise value-fill with the mask indicatorI (further details are in §4.4)." --> The second A should be M?
2. Can you add more comparisons against other established self-supervised learning frameworks for time series (e.g., TS2Vec, contrastive methods) using the same backbone?
3. Can you present an ablation study on whether the asymmetric augmentation strategy is necessary?
4. No hyperparameters are given (e.g., dimensions of the heads, the choice of alpha when balancing the two losses, masking ratio, noise levels, etc.).
5. The proposed framework seems to be very similar to many of the existing time series pretraining papers that use contrastive learning.

**Questions:**

1. Do TTM (Ekambaram et al., 2024) and TSPulse (Ekambaram et al., 2025) also use a subset of 1B time points drawn from the combined dataset of the Monash (Godahewa et al., 2021) and LibCity (Wang et al., 2021; Woo et al., 2024) data collections? I am confused by your writing in Section 5, Pre-training Setup and Training Dataset. What are the original training objectives for TTM and TSPulse?
2. Can you describe the experiment settings in Section 5.1.1 in more details?
3. Can you redraw Figure 1? What does the word "Centering" mean in this figure? In the last column, the visible patch for teacher should be b_T^{(p)}, while the reconstructed patch for student should be b_S^{(p)}? The figure currently looks very messy. I suggest to redesign it.
4. In Table 2, TSDINO shows clear gains in zero-shot tasks but is identical to the baseline when finetuned. Why do the representation improvements not translate to the fine-tuning setting?

---

> ### Author Response · Authors · 2025-12-04
> **Response to Reviewer dBGd**
>
> Thank you for the helpful comments and suggestions.
>
> > Writing and notation issues
>
> Thank you very much for your thoughtful suggestions. We rvised the manuscript accordingly.
>
> > The proposed framework seems to be very similar to many of the existing time series pretraining papers that use contrastive learning.
>
> The proposed method differs from contrastive learning in that it uses a teacher-student distillation approach rather than contrasting positive and negative pairs, while both methods aim to learn robust representations by leveraging augmentations of the input data. We discussed the differences between our method and contrastive learning in Section 1. Specifically, contrastive learning methods requires large negative sets; thus, it does not scale efficiently. In addition, our method incorporates masked patch prediction to capture local temporal structures, which is not commonly addressed in traditional contrastive learning frameworks.
>
> > Do TTM (Ekambaram et al., 2024) and TSPulse (Ekambaram et al., 2025) also use a subset of 1B time points drawn from the combined dataset of the Monash (Godahewa et al., 2021) and LibCity (Wang et al., 2021; Woo et al., 2024) data collections? I am confused by your writing in Section 5, Pre-training Setup and Training Dataset. What are the original training objectives for TTM and TSPulse?
>
> Yes, both TTM and TSPulse were pre-trained on the same subset of 1B time points drawn from the combined dataset of Monash and LibCity. The original training objective for TTM was to minimize the Mean Squared Error (MSE) between the predicted future values and the actual future values in a forecasting task. For TSPulse, the original training objective was to minimize the combination of reconstruction loss (via dual-space masked reconstruction in both time and frequency domains) and sequence semantics loss, which encourages disentangled embeddings for high-level representations.
>
> > Can you describe the experiment settings in Section 5.1.1 in more details?
>
> In Section 5.1.1, we evaluated the time-series search and retrieval performance of TSDINO using the UCR dataset, where we used the pre-trained models to extract embeddings for the time series data and then performed nearest neighbor search in the embedding space with cosine similarity as the distance metric.

---

### Meta-Review · Area_Chair_jVqf · 2025-12-22

**Summary:**

This paper introduces TSDINO, a teacher–student pretraining paradigm that leverages label-free self-distillation to enhance the robustness of time-series foundation models. The proposed self-supervised framework is empirically evaluated and demonstrates promising performance relative to selected methods.

**Reviewer Concerns:**

Most reviewers expressed significant concerns regarding both the presentation quality and the claimed contributions of the paper. In particular, multiple writing and notation issues were identified. Reviewer WNzk noted that it is somewhat overstated to claim that pre-training strategies have not been previously explored in the context of TSFMs. Reviewer umYb pointed out that the experimental evaluation is relatively weak, as it lacks validation on large-scale benchmark datasets. Reviewer aesc further emphasized that the conclusions drawn in the paper are not sufficiently supported by the presented evidence.

Moreover, the authors did not adequately address all of the concerns raised by the four reviewers in their rebuttal.

**Reviewer Scores:**

The authors did not fully address the reviewers’ concerns, and no discussion took place among the reviewers. The initial overall scores assigned by the four reviewers were 2, 4, 2, and 2. Based on the partial rebuttal provided, these evaluations are unlikely to be revised.

---

### Decision · Program_Chairs · 2026-01-26

Reject